# Study protocol for a systematic review with meta-analysis to compare digital stress tests regarding their psychological and physiological stress responses

**Lara Reiser, Luciana Diaconescu, Nicolas Rohleder***

Chair of Health Psychology, Institute of Psychology, Friedrich-Alexander-Universität Erlangen-Nürnberg, Erlangen, Bavaria, Germany

* nicolas.rohleder@fau.de

## Abstract

### Background and aim

Stress is a phenomenon of everyday life that all humans are exposed to, and that can have negative effects on health. Exposure to chronic stress has been associated with development of several diseases, but also acute stress can have detrimental health effects. To understand acute stress, standardized laboratory protocols are frequently employed. However, these protocols require considerable personal and financial resources, and are therefore usually uneconomical. More and more digitized, automated protocols are currently being developed, which may enable a more cost-effective and less time-consuming implementation, but these have not been systematically examined and compared. The aim of the proposed systematic review is therefore to provide a comprehensive overview of different digital stress tests and to compare them in terms of the elicited stress responses.

### Methods

This study protocol was registered with PROSPERO, and PRISMA-P guidelines were followed. We will screen several online databases for eligible studies in English or German without any restrictions on the publication date using an a priori defined search strategy. Additionally, we will search those papers for other eligible studies. The primary outcome will be stress and/or similar psychological states. The search, selection and data extraction processes will be conducted by two independent reviewers. In case of discrepancy a third reviewer will be consulted. If possible, we will perform meta-analyses. To determine confidence in results we will assess risk of bias and overall quality of evidence. Additionally, results will be critically examined to avoid meta-biases.

### Discussion

To the best of our knowledge, no systematic reviews or meta-analyses are published comparing digital stress tests - designed to elicit stress and/or similar psychological

**Data availability statement:** No datasets were generated or analysed during the current study. All relevant data from this study will be made available upon study completion.

**Funding:** The author(s) received no specific funding for this work.

**Competing interests:** The authors have declared that no competing interests exist.

states. In this study protocol, we propose a systematic review analyzing existing digital stress tests, and examining their effectiveness by measuring acute stress responses.

## Introduction

Navigating through our environment, we are frequently exposed to situations that can be evaluated as stressful, and therefore have a significant impact on our health. With regard to the health effects of stress, a distinction needs to be made between chronic and acute stress. Chronic stress is defined as long-term experiences of adverse psychosocial conditions, such as unemployment, occupational stress, difficult interpersonal relationships, or having to care for a family member with a chronic disease [1]. Chronic stress exposure has been shown to be a predictor of depression, cardiovascular disease, and other non-communicable diseases [2]. During acute stress, an experience is interpreted as a threat by the central nervous system (CNS). Consequently, both physiological responses and changes in behavior are initiated, leading to allostasis and adaptation. Orchestrated by the CNS, the body mounts a short-term stress response to provide energy, enabling individuals to cope with challenges [3]. Over time, repeated activation of stress systems can lead to accumulation of allostatic load, in which excessive exposure to mediators of endocrine, neural, and immune stress can have negative effects on various organ systems and thereby contribute to diseases [3].

Many stressors in modern societies are characterized by their repeated, acute nature, such as interruptions by digital technologies [4]. Therefore, in addition to investigating chronic stress, it is important to understand the individual response of human beings to acute stress, which is typically examined in laboratory settings using standardized stress protocols [1].

The considerable personal and financial resources required for these protocols limit their widespread use and impede a more comprehensive understanding of the health effects of acute stress. To address these disadvantages, new digitized and automated protocols are being developed, potentially allowing for more cost-effective and less time-consuming implementation. However, these new protocols have not yet been systematically investigated and compared. Consequently, the aim of this systematic review is to provide an overview of existing digital stress induction protocols, to compare them in terms of their potential to activate the two major physiological stress systems—the sympatho-adrenal-medullary (SAM) axis and the hypothalamic–pituitary–adrenal (HPA) axis—and to evaluate the psychological and physiological stress responses they elicit, based on reported effect sizes.

A number of laboratory protocols have been developed that all have the aim to induce acute stress in human research participants and under laboratory conditions. Among them, the Trier Social Stress Test (TSST; [5]) is considered the gold standard. It reliably activates the HPA axis, the sympathetic nervous system (SNS), and its SAM components [6,7]. It also affects further downstream systems, such as the cardiovascular, immune, and metabolic systems [8]. Several adaptations exist, including the group version (TSST-G; [9]), the child version (TSST-C; [10]), and non-stress control variants such as the friendly TSST (f-TSST; [11]) and the placebo TSST [12]. These modified versions differ mainly in their social-evaluative and uncontrollability components, resulting in weaker HPA axis activation while still stimulating the SNS. The TSST contributed significantly to developing a more comprehensive understanding of psychological and physiological acute stress responses and was also shown to be the most successful protocol for HPA axis activation [6].

Besides the TSST, a number of other established stress induction protocols have been developed, including the Trier Mental Challenge Test (TMCT; [13]), the Montreal

Imaging Stress Task (MIST; [14]), the Stroop Color-Word Interference Test [15], the Cold Pressor Test (CPT; [16]), the Socially Evaluated Cold-Pressor Test (SECPT; [17]), the Maastricht Acute Stress Test (MAST; [18]), the Paced Auditory Serial Addition Task (PASAT; [19]), the $CO_2$ challenge test [20], noise stress [21], and the Mannheim Multicomponent Stress Test (MMST; [22]). These protocols provide a comprehensive framework for examining the physiological, psychological, behavioral, emotional, and cognitive consequences of acute stress in humans [23].

While most of these protocols have been successfully used to elicit psychological and physiological stress responses, they also come with a number of disadvantages. One of the main limitations is their high demand for resources, including the availability of testing facilities, personnel capacity, and participant time, as well as the requirement for participants to travel to a laboratory setting instead of completing the test at home or in a work environment. Hence, most of these protocols are uneconomical and additionally require considerable personal and financial resources. Furthermore, studies that are carried out with the help of human research assistants could be burdened by experimenter effects [24]. Therefore, it appears to be essential in the context of future research to minimize the resource requirements. A promising solution to these problems could be electronic, virtual or otherwise automated stress protocols.

For better generalizability of results, it could also be beneficial to induce acute stress in the natural environment of the tested individuals in addition to the laboratory setting. This dual approach would enable more effective exploration of how surrounding environments influence both physiological and psychological stress response. It would also facilitate a clearer understanding of the causal relationships between environmental components and the physiological and psychological stress response.

While some automated stress protocols have been proposed and tested, it needs to be formally established whether automated stress protocols reliably elicit a psychological and/or physiological stress response. Questions also arise regarding whether these responses are comparable in intensity and duration to those induced by conventional stress protocols. Additionally, there is a need to identify which individual components of digital stress tests are particularly effective in eliciting stress responses in the tested participants.

A number of different automated and/or digitized versions of the TSST are already being used for a wide variety of research purposes. The virtual TSST (V-TSST) versions are completely digitized adaptations of the regular TSST by Kirschbaum et al. [5] using virtual reality (VR) settings such as 3D projections or immersive headgear.

In 2019, Helminen et al. [25] conducted a meta-analysis of 13 studies which utilized the V-TSST for stress induction. The authors measured cortisol levels before test administration and at peak stress. They found a medium average effect size (ES = 0.65) for the increase in cortisol from baseline to peak value. However, despite effectively inducing psychosocial stress according to this review, the cortisol response elicited by the V-TSST was lower compared to the conventional TSST [25]. In a systematic review with meta-analysis, Helminen et al. [26] conducted a comparison between studies utilizing V-TSST protocols and those employing the traditional TSST. According to the authors, studies using the V-TSST demonstrated stress (responses in cortisol, heart rate, self-reported stress) that were comparable to those observed in traditional TSST studies.

The TSST is not the only stress protocol which has been digitized. The goal of the Digital Stress Test (DST; [27]), a web-based protocol, was to provide a scalable digital tool which allows standardized induction and recording of acute stress responses outside of the laboratory without contact to an experimenter. The DST was developed on the basis of well-researched protocols for stress induction. Participants who completed the DST showed

significantly higher perceived stress indices, measured by visual analogue scales [28] as well as by the international Positive and Negative Affect Schedule Short Form [29], than the participants who were tested with a control condition of the DST on all post-baseline measurements. Similar effect sizes for negative affect were measurable in the participants who underwent the DST as in the participants who were tested in conventional TSST studies [27].

## Summary and aim of the systematic review

The aim of the proposed systematic review is to provide a comprehensive overview of different digital stress tests and to compare them with regard to their potential to activate the two main physiological stress systems—the SAM axis and the HPA axis—as well as their impact on self-reported stress levels. The ultimate goal is to help researchers develop and deploy digital stress tests, which are optimized to elicit the desired psychological and/or biological responses. To this end, the review will assess both psychological and physiological stress responses. By synthesizing effect sizes reported across studies, the review seeks to evaluate and compare the effectiveness of different digital stressors in eliciting stress responses across both axes as well as on the level of self-reported stress.

In the context of this review, a digital stress test is defined as any form of intervention or procedure designed to elicit stress and/or similar psychological states and that is conducted automatically and follows a predetermined and standardized protocol, either in a real-world setting or in a controlled environment, utilizing digital technologies such as VR environments, computers, smartphones, VR headsets, and similar equipment. While experimenters may engage with participants before and after the test execution, they must not intervene during the procedure itself, which must be executed entirely autonomously.

This review aims to identify which of the existing conventional stress protocols have already been adapted into digital formats, and which tests have been developed as digital procedures from the outset. In doing so, we will examine whether and to what extent these digital protocols are capable of eliciting stress responses—specifically regarding the activation of the SAM and/or HPA axis as well as changes in self-reported psychological stress. To ensure a broader and more informative evidence base, we will not restrict our analysis to protocols explicitly labeled as digital stress tests. Rather, we will also include studies investigating digital procedures designed to elicit stress and/or similar psychological states—such as increased negative affect or self-reported psychological strain—as long as these procedures follow a fully automated and standardized protocol and include adequate physiological and/or psychological outcome measures. This approach will enable us to derive insights into which types of digital stressors most reliably induce measurable stress responses across different modalities.

## Materials and methods

### Protocol registration and reporting information

This protocol has been registered within the PROSPERO database (registration ID: CRD42024555457). It is being reported in accordance with the reporting guidance provided in the Preferred Reporting Items for Systematic Review and Meta-analysis Protocols (PRISMA-P) statement [30,31] (see PRISMA-P checklist in S1 Table). The proposed systematic review will be reported in accordance with the reporting guidance provided in the Preferred Reporting Items for Systematic Reviews and Meta-analyses (PRISMA) statement [32,33].

### Eligibility criteria

Studies will be selected according to the following study characteristics: study design, participants, interventions, control/comparators, outcomes (PICO framework [34]):

**Study design:** We will include only studies employing fully automated and standardized digital protocols that are designed to elicit stress and/or similar psychological states— such as increased negative affect or psychological strain—and that incorporate adequate physiological and/or psychological measures to assess the respective stress responses, regardless of whether these protocols are explicitly labeled as digital stress tests. We expect these studies to be cross-sectional studies, in which a group of participants is exposed to the digital stress task, and in which the respective measures of stress are taken before and after the stress situation. These studies should ideally have between- or within-subject control conditions, but we will also include adequately designed studies in which only pre- and post-stress measurements are collected, and in which the pre-stress measurement serves as a baseline to quantify the respective stress response.

**Participants:** We will include all healthy adult participants (18 years or older) who have no known explicit somatic or psychiatric disorders. Specifically, individuals known to be undergoing chronic psychopharmacological treatment will not be included in the systematic review. We will further exclude studies with participants with any pharmacological treatment that is known to interfere with stress systems (e.g. glucocorticoids or beta-blockers). However, if studies involve healthy control groups (undergoing the same digital stress test) alongside psychiatric patients, these control groups will be included.

**Interventions:** We will include all kinds of interventions in the form of fully automated and standardized digital protocols that are designed to elicit a stress response—whether psychological and/or physiological—and that incorporate adequate measures to assess the respective responses. This includes protocols explicitly labeled as digital stress tests as well as those aiming to induce stress and/or similar psychological states, such as increased negative affect or perceived stress, provided they meet the definitional criteria for digital stress tests used in this review.

**Control/Comparison:** We will include all studies that ideally have between- or within-subject control conditions, but we will also include adequately designed studies in which only pre- and post-stress measurements are collected, and in which the pre-stress measurement serves as a baseline to quantify the respective stress response.

**Outcomes:** As psychological stress responses, we will accept any repeated (pre-/post-) measure, in which a self-report measure of stress or associated psychological construct, will be assessed at least twice, i.e. before and after the respective stress intervention, so that a change score can be computed. As physiological stress responses, we will accept any repeated (pre-/post-) measure of a physiological state that is associated with psychological stress, for example salivary cortisol, salivary alpha-amylase, electrophysiological measures, such as heart rate (variability), skin conductance, blood pressure, and blood-based biomarkers of stress. We will also include any studies that employ any kind of contactless measure of stress. For studies with adequate control conditions, we will use Cohen's *d* as effect size measure (for studies with small samples, we might also use Hedge's *g*). For studies in which only pre- and post-stress measurements are collected, we will use the standardized mean gain effect size as described by Lipsey and Wilson [35], in accordance for example with Helminen et al. [25]. We will select studies without any restrictions on publication dates and include articles in peer reviewed journals reported in the languages English and German. We will exclude conference proceedings, dissertations and theses.

## Information sources and search strategy

As a primary source of relevant studies, we will conduct a structured search in several electronic databases: PubMed/MEDLINE, PsycINFO, Web of Science, Scopus and Cochrane Central Register of Controlled Trials (CENTRAL). The secondary source for identifying additional eligible papers will be a forward and backward citation analysis using the tool SpiderCite from the software Systematic Review Accelerator [36], searching the selected studies for other suitable papers. The review team will perform the literature search which will include a broad range of keywords and terms related to the PICO framework (e.g., "digital", "stress", "TSST", "MIST", "cortisol", "heart rate variability", "negative affect"). In S2 Table, a full draft of the search strings for PubMed/MEDLINE is provided. This search strategy was initially adapted to the other databases using the Systematic Review Accelerator [36], which facilitated the transfer of syntax. However, we are aware that the Systematic Review Accelerator does not translate subject headings or controlled vocabulary (e.g., MeSH in PubMed). Therefore, we manually reviewed and adjusted all controlled vocabulary terms for each database to ensure conceptual consistency and avoid inaccurate or overly broad mappings. The final search strings reflect these manual refinements and can be found in S3 Table. The search process itself will be conducted by two independent reviewers (LR, LD), with discrepancies resolved by a third reviewer (NR) to enhance reproducibility and reduce bias.

## Screening and selection procedure

All retrieved titles and abstracts of identified articles will be imported into the software Paperpile (Paperpile LLC). The screening process will be conducted using the web and mobile application Rayyan [37]. A stepwise and systematic selection of eligible studies, that is, screening of titles, abstracts, and full texts, will be conducted by two independent reviewers (LR, LD). Until a consensus is reached the potential discrepancy will be discussed by the reviewer team. Excluded studies will be recorded. For the purpose of identifying potential misunderstandings in the review team related to the eligibility criteria, a prior testing of all steps of the study selection will be conducted.

## Data collection process

The review team (LR, LD) will extract the data and transfer it into Excel (Microsoft 365). To test for comprehensiveness and feasibility this step will be pre-tested with five articles and in case of diverse opinions and disagreement a third reviewer (NR) will be involved. From all selected articles individual data and several main categories will be retrieved. The corresponding authors will be contacted if information is missing to obtain relevant data.

## Data processing and classification of exposure and outcome variables

Extracted data will include information on title of the paper, year of publication, author(s), sample size, country of recruitment, inclusion criteria, number of and reasons for exclusions, final number of participants, mean age, standard deviation of age, difference in percentage between female and male participants, sample type (ordinary experimental groups vs. healthy control groups undergoing the same procedure alongside psychiatric patients), measure(s) of stress (psychological vs. physiological), description of the stress intervention and control intervention, study setting (laboratory vs. field), used technology (i.e. headgear, virtual environment), role of experimenter, order of conditions (in case of within-subject design), and mean and standard deviation of used stress metric at pre-assessment and post-assessment

for the experimental group and control group. Methodological quality of considered studies will be assessed. In case of high heterogeneity and an adequate number of studies, the eligible studies will be categorized with regard to the conducted digitized stress protocol, e.g. digitized versions of TSST [25], TMCT [13], Stroop [15]. Furthermore, the selected studies will be classified into physiological and psychological parameters with regard to the outcome variables.

## Risk of bias in individual studies

To assess the methodological quality of all eligible studies and identify any potential limitations to validity, a standardized risk of bias evaluation will be conducted. To correctly review the diverse study designs, two different established tools will be used in order to minimize the risk of bias. The Cochrane Risk of Bias Tool (RoB 2; [38]) will be applied for randomized controlled trials and for non-randomized studies of interventions, the Risk Of Bias In Non-randomized Studies – of Interventions (ROBINS-I; [39]) tool will be used. These two risk of bias assessment tools feature sets of questions that cover various domains of potential bias, ranging from selection to reporting, and require responses based on specific judgments. A third reviewer (NR) will be involved in case of disagreement between the two reviewers (LR, LD).

## Data synthesis

If an adequate number of high-quality studies with relatively low heterogeneity are obtained, we will quantitatively synthesize data from primary studies through a meta-analysis. As part of the meta-analyses, we will compare effect size measures (Cohen's *d* or Hedge's *g*) between studies with different digital stress paradigms. If no control conditions are available, we will compare the standardized mean gain effect size [35] between studies using different digital protocols. If meta-analysis is not possible, presentation of results in narrative and tabular format will be provided. If substantial heterogeneity and inappropriateness of statistical pooling appears, graphical summary approaches for evidence synthesis in the absence of meta-analysis, i.e., effect directions, harvest plots, or bubble plots to summarize information in a user-friendly and accessible manner (see [40]) will be applied. If an adequate number of high-quality studies with low heterogeneity will be found, extracted data from primary studies will be quantitatively synthesized in a meta-analysis, using R 4.4.1 (package: metafor [41]). In case of relatively high heterogeneity of effects in individual studies, we will select a random effects model based on the DerSimonian and Laird [42] method, in order to estimate the average of the effects across studies. Heterogeneity will be evaluated by using Cochran's *Q* test [43] and $I^2$ statistic [44], estimating the variance between individual studies. We will illustrate results graphically using forest plots including individual study effects (step 1), combined effect estimates (step 2) and confidence intervals, respectively [45].

**Additional analyses.** If the number of studies and heterogeneity in study populations allow it, subgroup analyses concerning specific characteristics within our eligibility criteria (i.e., participants' demographics, stressors, digital stress tests, outcomes, study designs and quality) will be performed.

**Meta-biases.** We will examine the results of this review (if possible) with meta-analyses for meta-biases, such as publication bias across studies or selective reporting within studies. Tests for asymmetry including a minimum of ten studies (if possible), such as Egger's test [46], and funnel plots will be conducted to check for small-study effects [47,48].

**Confidence in cumulative evidence.** The strength of the evidence base will be estimated by using the Grading of Recommendations Assessment, Development and Evaluation

(GRADE) system, a framework for rating the quality of evidence and strength of recommendations [49,50]. Quality of evidence refers to the certainty that the estimates of the effect are valid and can be ranked into high, moderate, low, and very low [50,51]. Quality rating can be downgraded, if the five factors (study limitations, inconsistent results, indirectness of evidence, imprecision and publication bias) are present [52]. The strength of recommendation is classified by the confidence that preferred effects outweigh the unpreferred effects and can be categorized as strong or weak [50,53].

## Discussion

In this study protocol, we propose a systematic review that will analyze existing digital stress tests—defined as fully automated and standardized procedures utilizing digital technologies to elicit stress and/or similar psychological states—by examining their effectiveness in triggering acute stress responses, assessed through physiological and/or psychological parameters. Specifically, we will focus on the potential of these digital protocols to activate the two main physiological stress systems, the SAM axis and the HPA axis, as well as to induce self-reported stress or related psychological responses such as negative affect.

As a first step, we will screen major web databases, including PubMed/MEDLINE, PsycINFO, Web of Science, Scopus and CENTRAL, for studies proposing or examining digital stress tests, as well as digitized versions of established conventional stress tests. While previous meta-analyses have addressed specific digital protocols such as the V-TSST and its variations [25,26], to our knowledge, no comprehensive systematic review or meta-analysis has yet compared all types of digital stress tests designed to elicit stress and/or similar psychological states autonomously in digital environments without experimenter intervention.

For the purpose of this review, digital stress tests will be defined as interventions conducted automatically according to standardized, predetermined protocols in either controlled or real-world settings using digital technologies such as virtual reality environments, computers, smartphones, or similar devices. Although experimenters may engage with participants before and after the test, the stress induction itself must proceed entirely autonomously, without experimenter interference.

Our systematic review aims to provide a detailed overview of which conventional stress protocols have been successfully digitized and which have been developed as fully digital from the outset. Moreover, we will examine to what extent these digital protocols effectively elicit stress responses, specifically regarding activation of the SAM and/or HPA axis and self-reported psychological stress. By synthesizing effect sizes across studies, we intend to evaluate and compare the relative effectiveness of these digital stressors on both physiological and psychological levels.

Findings from this systematic review will be disseminated through publication in a peer-reviewed open-access journal and presentation at relevant academic conferences. A preprint will be posted to ensure early accessibility. All supplementary materials—including data extraction templates and any analysis scripts—will be publicly available via our preregistration project on the Open Science Framework (OSF). This repository will be updated with data and additional documentation at the time of publication.

## Supporting information

**S1 Table. PRISMA-P 2015 Checklist.**
(PDF)

**S2 Table. Search strings structured by the PICO framework for PubMed.**
(PDF)

**S3 Table. Combined search strings for all databases.**
(PDF)

## Author contributions

**Conceptualization:** Lara Reiser, Luciana Diaconescu, Nicolas Rohleder.

**Investigation:** Lara Reiser, Luciana Diaconescu, Nicolas Rohleder.

**Methodology:** Lara Reiser, Luciana Diaconescu.

**Supervision:** Nicolas Rohleder.

**Writing – original draft:** Lara Reiser, Luciana Diaconescu.

**Writing – review & editing:** Lara Reiser, Luciana Diaconescu, Nicolas Rohleder.

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
