## [Decision Letter · Decision Letter 0]

29 Apr 2025

PONE-D-24-53797Study protocol for a systematic review with meta-analysis to compare digital stress tests regarding their psychological and physiological stress responsesPLOS ONE

Dear Dr. Rohleder

Thank you for submitting your manuscript to PLOS ONE. After careful consideration, we feel that it has merit but does not fully meet PLOS ONE’s publication criteria as it currently stands. Therefore, we invite you to submit a revised version of the manuscript that addresses the points raised during the review process. Please submit your revised manuscript by  Jun 13 2025 11:59PM. If you will need more time than this to complete your revisions, please reply to this message or contact the journal office at plosone@plos.org. Please include the following items when submitting your revised manuscript:

We look forward to receiving your revised manuscript.

Kind regards,

Efrem Kentiba, PhD

Academic Editor

PLOS ONE

Journal Requirements:

Reviewers' comments:

Reviewer's Responses to Questions

**Comments to the Author**

1. Does the manuscript provide a valid rationale for the proposed study, with clearly identified and justified research questions?

Reviewer #1: Yes

Reviewer #2: Yes

Reviewer #3: Yes

2. Is the protocol technically sound and planned in a manner that will lead to a meaningful outcome and allow testing the stated hypotheses?

Reviewer #1: Partly

Reviewer #2: Yes

Reviewer #3: Yes

3. Is the methodology feasible and described in sufficient detail to allow the work to be replicable?

Reviewer #1: Yes

Reviewer #2: Yes

Reviewer #3: Yes

4. Have the authors described where all data underlying the findings will be made available when the study is complete?

Reviewer #1: No

Reviewer #2: No

Reviewer #3: No

5. Is the manuscript presented in an intelligible fashion and written in standard English?

Reviewer #1: Yes

Reviewer #2: Yes

Reviewer #3: Yes

6. Review Comments to the Author

You may also provide optional suggestions and comments to authors that they might find helpful in planning their study.

Reviewer #1: As I am an academic librarian I am mostly confining my thoughts to the data collection/search strategy stage.

GENERAL COMMENTS

This paper intends to map the literature on virtual stress testing and to compare the responses provided by different tests. It is therefore not a systematic review but a scoping review. I encourage the authors to consult guides on this type of knowledge synthesis.

This change of review type has implications for the nature of the search and for the structure of the final paper.

THE SEARCH

The search strategy provided in Appendix 2 is confusingly presented, has multiple flaws, and does not appear to have been constructed logically and with clear separation of concepts. It would be very helpful if the authors could just provide the final search string in a clear way - to answer the question "What are you actually going to type into PubMed"?

For example, "Stress OR social OR reactivity OR arithmetic OR induction OR public OR speech OR strain OR challenge OR MIST OR stroop OR TSST OR social-evaluative OR threat OR psychosocial OR virtual" combines with OR a variety of different concepts, all of which are thus, of course, alternatives. Why are "virtual" and "stress" both alternatives when they are different concepts?

SEARCH-WIDE COMMENTS

1. SYSTEMATIC REVIEW ACCELERATOR: While this is a useful tool, note that it only translates SYNTAX and does not separately translate subject headings/controlled vocabulary such as between MeSH/Emtree. This is very important because automatic term mapping between MeSH/Emtree can lead to vastly overinflated search results which will be highly time-consiuming to search. The authors will also need to include a degree of manual translation.

2. CHOSEN DATABASES:

* Google Scholar. This certainly has its uses in knowledge synthesis, but clearly does not provide a discrete body of literature than can be downloaded into Rayyan. Will the authors be screening at the search stage? How will they translate their search? How will they ensure reproducibility? What is the strategy here? I strongly recommend removing this database, or noting that it was used in early stages to help with seed articles for other searches.

* CINAHL or Emcare would have been a good choice of additional database.

3. OTHER COMMENTS

* OUTCOMES: Why are you excluding conference proceedings? Part of the rationale for a scoping review is to map the state of the art of a particular field. What if a conference proceedings leads you to a later journal article you would otherwise have missed? It is particularly odd considering that you are happy to include clinical trials, presumably including incomplete trials.

* Have you deliberately decided against hand screening and grey literature searching?

* I don't understand "...which are based on an unsystematic literature search" (bottom p12). Did you start with an unsystematic search and then improve it? Or am I reviewing an unsystematic search?

OTHER COMMENTS ON SPECIFIC CONCEPT STRATEGIES

* Is the final search a combination of the sections starting ""humans"[MeSH Terms] NOT...", "digitalisation"[All Fields]..." and "stress"[All Fields]", or is it the combined search provided at the bottom?

* If it is the former, then please see the comments below.

* If it is the latter, then:

* why have all the wildcards been removed?

* why have you removed most of the subject headings?

* why are you combining "virtual" with "stress" using an OR?

* "Humans" concept: This is poorly constructed and has consequences for the search which I am sure are unintended.

* By using a NOT, the authors are removing not only articles discussing just infants etc, but also any articles discussing those ALONG WITH adults. It is a highly dangerous option and is guaranteed to remove relevant articles.

* By combining this concept with AND, the authors are removing all unindexed articles as the concept is defined using only subject headings. As articles are not always indexed quickly this will remove a significant number of recent articles of relevance. I am sure the authors did not intend for either of these consequences.

* The authors should instead employ one of the many available hedges for age group limitation, or consult an information professional/librarian with expertise in these searches.

* "Virtual/digital stress tests" concept: Clearly this is a combination of the concepts of "virtual/remote/telehealth" and "stress testing".

* "Virtual" subconcept:

* Why is the term "virtual" in the "stress" concept?

* Starting with "digital*[all fields] or digitiz*[all fields] or digitis*[all fields]" would be more concise and ensure that all terms include spellings with "s" and "z", which the authors fail to do. See https://pubmed.ncbi.nlm.nih.gov/help/#wildcards.

* For smartphones I recommend instead "exp Computers, Handheld/ OR exp Cell Phone/"

* Multiple other issues such as not including "smart phone", "smart device" etc (spaces, plus synonyms) and missing terms such as "telehealth". A check of previous systematic reviews on these subjects should have been conducted to mine appropriate search terms.

* "Stress" concept:

* The presence of terms like "stroop s"[all fields] demonstrates that the authors created this search using Automatic Term Mapping rather than actually considering the terms they used individually. This is very much not recommended for knowledge synthesis projects as it leads to searches lacking in rigour, the tool is mainly aimed at helping the lay public with searches.

* "social*"[all terms] would remove most of the noise from this search, see above.

* The authors spend much time in their excellent introduction discussing tests like V-TSST and then fail to include them in the search. Why?

* "Social behavior" is way too broad a MeSH term. Why not "Stress, Psychological"?

* Consider also "exp Psychological Tests/" combined with various stress/anxiety terms.

* Much of the remainder of the strategy for this concept OR's concepts like "psychosocial OR virtual", surely these should be AND'ed. Aren't both necessary?

FINAL RECOMMENDATIONS

In any other scientific study great care would be taken to ensure the quality of the data, and to work with an expert who can ensure that quality. Knowledge syntheses are no exception. I strongly urge the authors to engage the services of one of the librarians at their university library to assist with their search and to improve its quality, taking into account the observations above. Note that these observations are not complete and are intended to guide the authors into more wide-ranging improvements.

Because this search is inadequate, meaning that the resulting conclusions will be build on a poor foundation, I have recommended a major review. This is necessary to tackle the shortcomings of the search and ensure great rigour and reliability in the final product.

Reviewer #2: Dear Authors,

Reviewing the protocol for a new review is always an exciting endeavor, and I agree that it is a necessary step to ensure an evidence-based outcome. I would like to draw your attention to a few discussion points in your manuscript, and I hope these remarks prove helpful for your future work:

According to the statements in the text, it is unclear whether the provided draft search string is subject to further revisions or represents the final outcome. The material appears to be presented in future tense, yet the search string seems to have already been finalized.

It would be advisable to specify the study designs that will be included (e.g., RCTs, non-RCTs, etc.).

Do you plan to review grey literature? Additionally, are there any language restrictions for the studies (for example, studies written only in English)?

It might be beneficial to provide a clearer classification regarding the primary outcome of the publication. I am confident that you have a plan for this, but a more explicit description would enhance clarity.

I hope these comments will be useful as you refine your manuscript.

Reviewer #3: Overall, the manuscript is interesting and outlines a protocol for a meta-analysis or systematic review aiming to provide a comprehensive overview of various digital stress tests and to compare the elicited stress responses on both psychological and physiological levels. However, the Introduction is quite lengthy, spanning nearly seven pages. While the authors provide a thorough background, they introduce detailed descriptions of the different protocols too early. I recommend moving this detailed information to the Methods section. Additionally, the advantages and disadvantages of each protocol would be more appropriately discussed in the Discussion section. Overall, both the Introduction and the Discussion sections need to be substantially reworked, and this is the main issue with the paper in its current form.

7. PLOS authors have the option to publish the peer review history of their article (what does this mean?). If published, this will include your full peer review and any attached files.

Reviewer #1: **Yes: **Martin Morris

Reviewer #2: **Yes: **Iuliia Pavlova

Reviewer #3: **Yes: **Antonia Kaltsatou

---

## [Author Response · Author response to Decision Letter 1]

23 Jun 2025

please note that a formatted version has been uploaded with the revised files.

Point-by-Point Response to Reviewer Comments

Study protocol for a systematic review with meta-analysis to compare digital stress tests regarding their psychological and physiological stress responses

Lara Reiser, Luciana Diaconescu, Nicolas Rohleder

1. 1 General Response

We thank the Reviewers for their constructive comments that helped us to significantly improve our paper. We have addressed each comment as outlined below. The revised paragraphs in the manuscript are marked in blue and red, and additionally added to each specific reviewer’s comment below. Please note that a “clean” version of the manuscript is also included in our revision.

1. 2 Point-by-Point Response

1. 2.1 Response to Questions 1 to 5

1. 2.1.1 Q1: Does the manuscript provide a valid rationale for the proposed study, with clearly identified and justified research questions?

• • 3x “YES”; Response: thank you!

1. 2.1.2 Q2: Is the protocol technically sound and planned in a manner that will lead to a meaningful outcome and allow testing the stated hypotheses?

• • Reviewer #1: “partly”: addressed in response to specific comments below

• • 2x “YES”; Response: thank you!

1. 2.1.3 Q3: Is the methodology feasible and described in sufficient detail to allow the work to be replicable?

• • 3x “YES”; Response: thank you!

1. 2.1.4 Q4: Have the authors described where all data underlying the findings will be made available when the study is complete?

• • 3x “NO”; Response: We have decided to make results of our search publicly available in a public repository; we will use OSF for that.

1. 2.1.5 Q5: Is the manuscript presented in an intelligible fashion and written in standard English?

• • 3x “YES”; Response: thank you!

1. 2.2 Response to Reviewer #1

As I am an academic librarian, I am mostly confining my thoughts to the data collection/search strategy stage.

GENERAL COMMENTS

1. 2.2.1 This paper intends to map the literature on virtual stress testing and to compare the responses provided by different tests. It is therefore not a systematic review but a scoping review. I encourage the authors to consult guides on this type of knowledge synthesis.  This change of review type has implications for the nature of the search and for the structure of the final paper.

Response: We thank the Reviewer for the evaluation of our work and the helpful comments. After careful re-evaluation of our review’s aims, and of our manuscript, we have come to the conclusion that we are in fact aiming on going beyond the scope of a scoping review, and therefore, chose to continue with our initial plan on conducting a systematic review. We realize that we have not made our specific aims clear enough in the manuscript, and are therefore now providing clearer explanations in the relevant sections of our manuscript.

In response to the reviewer’s comment, we would like to clarify that the aim of our systematic review is to synthesize and compare the effectiveness of various digital paradigms in eliciting acute stress responses. Specifically, we assess both psychological and physiological indicators of stress in order to determine which digital stressors reliably activate the sympathetic-adrenal-medullary (SAM) axis, the hypothalamic-pituitary-adrenal (HPA) axis, and self-reported stress. By comparing the effect sizes of the reported stress responses, we aim to draw meaningful conclusions about the relative efficacy of different digital stress paradigms. Elaborating further, we intend to conduct a meta-analysis if the available data allow; at the very least, we will systematically summarize and compare the effect sizes of different stress induction protocols across various biomarkers and/or self-reported stress or related constructs. Importantly, we do not limit our analysis to paradigms explicitly labeled as digital stress tests. We also include digital protocols that may not have been primarily designed to induce stress but are nonetheless capable of eliciting acute stress responses or relevant psychological states associated with stress.

We have made a number of changes, the most important of which are highlighted below:

• - Introduction, pages 5 - 6, lines 105 – 135: “The aim of the proposed systematic review is to provide a comprehensive overview of different digital stress tests and to compare them with regard to their potential to activate the two main physiological stress systems — the SAM axis and the HPA axis — as well as their impact on self-reported stress levels. The ultimate goal is to help researchers develop and deploy digital stress tests, which are optimized to elicit the desired psychological and/or biological responses. To this end, the review will assess both psychological and physiological stress responses. By synthesizing effect sizes reported across studies, the review seeks to evaluate and compare the effectiveness of different digital stressors in eliciting stress responses across both axes as well as on the level of self-reported stress.

In the context of this review, a digital stress test is defined as any form of intervention or procedure designed to elicit stress and/or similar psychological states and that is conducted automatically and follows a predetermined and standardized protocol, either in a real-world setting or in a controlled environment, utilizing digital technologies such as VR environments, computers, smartphones, VR headsets, and similar equipment. While experimenters may engage with participants before and after the test execution, they must not intervene during the procedure itself, which must be executed entirely autonomously.

This review aims to identify which of the existing conventional stress protocols have already been adapted into digital formats, and which tests have been developed as digital procedures from the outset. In doing so, we will examine whether and to what extent these digital protocols are capable of eliciting stress responses — specifically regarding the activation of the SAM and/or HPA axis as well as changes in self-reported psychological stress. To ensure a broader and more informative evidence base, we will not restrict our analysis to protocols explicitly labeled as digital stress tests. Rather, we will also include studies investigating digital procedures designed to elicit stress and/or similar psychological states — such as increased negative affect or self-reported psychological strain — as long as these procedures follow a fully automated and standardized protocol and include adequate physiological and/or psychological outcome measures. This approach will enable us to derive insights into which types of digital stressors most reliably induce measurable stress responses across different modalities.”

1. 2.2.2 THE SEARCH  The search strategy provided in Appendix 2 is confusingly presented, has multiple flaws, and does not appear to have been constructed logically and with clear separation of concepts. It would be very helpful if the authors could just provide the final search string in a clear way - to answer the question "What are you actually going to type into PubMed"?  For example, "Stress OR social OR reactivity OR arithmetic OR induction OR public OR speech OR strain OR challenge OR MIST OR stroop OR TSST OR social-evaluative OR threat OR psychosocial OR virtual" combines with OR a variety of different concepts, all of which are thus, of course, alternatives. Why are "virtual" and "stress" both alternatives when they are different concepts?

Response: We thank the reviewer for this important comment. In response, we have revised our entire search strategy to improve conceptual clarity and ensure a more logical structure, with the aim of obtaining more relevant and appropriate results for our screening process. We have now separated key concepts and restructured the combinations accordingly. To increase transparency and reproducibility, we have created an additional file titled “S3 Table. Combined search strings for all databases”, which includes the complete and final search strings for each database we used. This file provides a clear overview of exactly what was entered into each search interface, including PubMed. We hope this revised documentation addresses the reviewer’s concerns.

1. 2.2.3 SEARCH-WIDE COMMENTS 1. SYSTEMATIC REVIEW ACCELERATOR: While this is a useful tool, note that it only translates SYNTAX and does not separately translate subject headings/controlled vocabulary such as between MeSH/Emtree. This is very important because automatic term mapping between MeSH/Emtree can lead to vastly overinflated search results which will be highly time-consuming to search. The authors will also need to include a degree of manual translation.

Response: We thank the reviewer for pointing out the limitations of using the Systematic Review Accelerator (SRA) for translating search strategies across databases. We agree that relying solely on automatic syntax translation can lead to conceptual inconsistencies, especially regarding controlled vocabularies such as MeSH (PubMed) and Emtree (Embase). In response, we have not only used the SRA to support the technical transfer of search syntax but have also manually reviewed and refined all subject headings to ensure that database-specific controlled vocabulary terms are appropriately selected and applied. This manual translation step was crucial to maintaining conceptual accuracy and avoiding overly broad or imprecise search results. The final, adapted search strategies for each database are included in the supplementary file “S3 Table. Combined search strings for all databases”. We hope this approach addresses the reviewer’s concerns regarding search precision and methodological rigor.

The following changes have been made to the manuscript:

Page 9, lines 209 - 214: “This search strategy was initially adapted to the other databases using the Systematic Review Accelerator [36], which facilitated the transfer of syntax. However, we are aware that the Systematic Review Accelerator does not translate subject headings or controlled vocabulary (e.g., MeSH in PubMed). Therefore, we manually reviewed and adjusted all controlled vocabulary terms for each database to ensure conceptual consistency and avoid inaccurate or overly broad mappings.”

1. 2.2.4 2. CHOSEN DATABASES: * Google Scholar. This certainly has its uses in knowledge synthesis, but clearly does not provide a discrete body of literature than can be downloaded into Rayyan. Will the authors be screening at the search stage? How will they translate their search? How will they ensure reproducibility? What is the strategy here? I strongly recommend removing this database, or noting that it was used in early stages to help with seed articles for other searches. * CINAHL or Emcare would have been a good choice of additional database.)?

Response: We thank the reviewer for this valuable comment regarding our choice of databases. Based on similar considerations, we have decided not to include Google Scholar in our final search strategy, as it does not allow for a clearly defined and reproducible body of literature to be exported and screened (e.g., via Rayyan). Regarding the reviewer’s suggestion to include CINAHL or Emcare, we carefully reviewed both databases. However, they primarily index literature focused on nursing and clinical practice, which does not align with the specific scope of our review on digital stress induction in experimental research settings. Therefore, these databases did not appear to add relevant studies for our research question. Instead, we decided to include Scopus as an alternative for Google Scholar. We hope this clarifies our rationale for database selection.

The manuscript has been updated as follows:

Page 9, lines 200 - 202: “As a primary source of relevant studies, we will conduct a structured search in several electronic databases: PubMed/MEDLINE, PsycINFO, Web of Science, Scopus and Cochrane Central Register of Controlled Trials (CENTRAL).”

1. 2.2.5 3. OTHER COMMENTS * OUTCOMES: Why are you excluding conference proceedings? Part of the rationale for a scoping review is to map the state of the art of a particular field. What if a conference proceedings leads you to a later journal article you would otherwise have missed? It is particularly odd considering that you are happy to include clinical trials, presumably including incomplete trials. * Have you deliberately decided against hand screening and grey literature searching? * I don't understand "...which are based on an unsystematic literature search" (bottom p12). Did you start with an unsystematic search and then improve it? Or am I reviewing an unsystematic search?.

Response: We thank the reviewer for raising these important points regarding our inclusion criteria and search strategy. First, regarding conference proceedings we fully agree that they can provide valuable insights, especially in emerging fields. However, we decided to exclude them in our systematic review due to the frequently limited information they provide on methodology, outcome measures, and results – factors that are essential for the planned extraction and synthesis of effect sizes. Second, concerning grey literature and hand searching we initially focused on peer-reviewed journal articles to ensure a high standard of methodological reporting and data quality. However, in accordance to the reviewer’s suggestion, we will conduct backward and forward citation tracking (snowballing) of included studies to minimize the risk of missing relevant publications as outlined in page 9, lines 202 - 205. Furthermore, we apologize for the lack of clarity in our wording concerning the literature search. The sentence in question was misleading and does not accurately reflect our methodological approach, which was based on a systematic literature search. To prevent any further confusion, we have removed the respective sentence from the manuscript.

OTHER COMMENTS ON SPECIFIC CONCEPT STRATEGIES

1. 2.2.6 OTHER COMMENTs #1 * Is the final search a combination of the sections starting ""humans"[MeSH Terms] NOT...", "digitalisation"[All Fields]..." and "stress"[All Fields]", or is it the combined search provided at the bottom? * If it is the former, then please see the comments below. * If it is the latter, then: * why have all the wildcards been removed? * why have you removed most of the subject headings? * why are you combining "virtual" with "stress" using an OR?

Response: Thank you for the insightful feedback. It helped us recognize that the provided file did not present our search strategy with sufficient precision.

To clarify, the final search strategy provided at the bottom is indeed the one we inserted into the PubMed search bar. In the supplementary table, we structured and categorized the overall search string using the PICO framework. In each cell, we first listed the conceptual search terms and then included the actual search string as processed by PubMed.

Regarding the specific points:

• • Wildcards and subject headings: While it may appear that wildcards and subject headings were removed, this is a result of how PubMed automatically processes search terms. For instance, when we entered the term "digital," PubMed automatically included variations and subject headings such as "digitalisation"[All Fields] OR "digitalised"[All Fields] OR "digitalization"[All Fields] OR…”. As a result, although wildcards and subject headings may not be explicitly displayed in the search string, they are functionally included in the search execution.

• • Combination of “virtual” and “stress”: We appreciate the reviewer’s observation. Initially, we had combined these terms using an OR operator, which indeed was not appropriate for the intended logic. We have now revised this section of the search to use an AND operator instead, in line with the reviewer's suggestion.

We have updated the search string accordingly and reflected these changes in the revised supplementary file named “S2 Table. Search strings structured by the PICO framework for PubMed”.

1. 2.2.7 OTHER COMMENTs #2 * "Humans" concept: This is poorly constructed and has consequences for the search which I am sure are unintend

---

## [Decision Letter · Decision Letter 1]

25 Aug 2025

Study protocol for a systematic review with meta-analysis to compare digital stress tests regarding their psychological and physiological stress responses

PONE-D-24-53797R1

Dear Dr. Rohleder,

We’re pleased to inform you that your manuscript has been judged scientifically suitable for publication and will be formally accepted for publication once it meets all outstanding technical requirements.

Kind regards,

Efrem Kentiba, PhD

Academic Editor

PLOS ONE

Additional Editor Comments (optional):

Reviewers' comments:

Reviewer's Responses to Questions

**Comments to the Author**

1. Does the manuscript provide a valid rationale for the proposed study, with clearly identified and justified research questions?

Reviewer #2: Yes

2. Is the protocol technically sound and planned in a manner that will lead to a meaningful outcome and allow testing the stated hypotheses?

Reviewer #2: Yes

3. Is the methodology feasible and described in sufficient detail to allow the work to be replicable?

Reviewer #2: Yes

4. Have the authors described where all data underlying the findings will be made available when the study is complete?

Reviewer #2: Yes

5. Is the manuscript presented in an intelligible fashion and written in standard English?

Reviewer #2: Yes

6. Review Comments to the Author

You may also provide optional suggestions and comments to authors that they might find helpful in planning their study.

Reviewer #2: Dear Authors, thank you for detailed answers. I do not have further comments or corrections. Kind regards,

7. PLOS authors have the option to publish the peer review history of their article (what does this mean?). If published, this will include your full peer review and any attached files.

Reviewer #2: **Yes: **Iuliia Pavlova

---

## [Editor Report · Acceptance letter]

PONE-D-24-53797R1

PLOS ONE

Dear Dr. Rohleder,

I'm pleased to inform you that your manuscript has been deemed suitable for publication in PLOS ONE. Congratulations! Your manuscript is now being handed over to our production team.

Kind regards,

on behalf of

Dr. Efrem Kentiba

Academic Editor

PLOS ONE